# On Fast Sampling of Diffusion Probabilistic Models

**Zhifeng Kong** [1]  **Wei Ping** [2]

## Abstract

In this work, we propose FastDPM, a unified framework for fast sampling in diffusion probabilistic models. FastDPM generalizes previous methods and gives rise to new algorithms with improved sample quality. We systematically investigate the fast sampling methods under this framework across different domains, on different datasets, and with different amount of conditional information provided for generation. We find the performance of a particular method depends on data domains (e.g., image or audio), the trade-off between sampling speed and sample quality, and the amount of conditional information. We further provide insights and recipes on the choice of methods for practitioners.

## 1. Introduction

Diffusion probabilistic models are a class of deep generative models that use Markov chains to gradually transform between a simple distribution (e.g., isotropic Gaussian) and the complex data distribution (Sohl-Dickstein et al., 2015; Ho et al., 2020). Most recently, these models have obtained the state-of-the-art results in several important domains, including image synthesis (Ho et al., 2020; Song et al., 2020b; Dhariwal & Nichol, 2021), audio synthesis (Kong et al., 2020b; Chen et al., 2020), and 3-D point cloud generation (Luo & Hu, 2021; Zhou et al., 2021). We will use "diffusion models" as shorthand to refer to these models.

Diffusion models usually comprise: *i)* a parameter-free $T$-step Markov chain named the *diffusion process*, which gradually adds random noise into the data, and *ii)* a parameterized $T$-step Markov chain called the *reverse* or *denoising process*, which removes the added noise as a denoising function. The likelihood in diffusion models is intractable, but they can be efficiently trained by optimizing a variant of

---

[1]UC San Diego, La Jolla, CA, USA [2]NVIDIA, Santa Clara, CA, USA. Correspondence to: Zhifeng Kong <z4kong@eng.ucsd.edu>, Wei Ping <wping@nvidia.com>.

Third workshop on *Invertible Neural Networks, Normalizing Flows, and Explicit Likelihood Models* (ICML 2021). Copyright 2021 by the author(s).

the variational lower bound. In particular, Ho et al. (2020) propose a certain parameterization called the denoising diffusion probabilistic model (DDPM) and show its connection with denoising score matching (Song & Ermon, 2019), so the reverse process can be viewed as sampling from a score-based model using Langevin dynamics. DDPM can produce high-fidelity samples reliably with large model capacity and outperforms the state-of-the-art models in image and audio domains (Dhariwal & Nichol, 2021; Kong et al., 2020b). However, a noticeable limitation of diffusion models is their expensive denoising or sampling process. For example, DDPM requires a Markov chain with $T = 1000$ steps to generate high quality image samples (Ho et al., 2020), and DiffWave requires $T = 200$ to obtain high-fidelity audio synthesis (Kong et al., 2020b). In other words, one has to run the forward-pass of the neural network $T$ times to generate a sample, which is much slower than the state-of-the-art GANs or flow-based models for image and audio synthesis (e.g., Karras et al., 2020; Kingma & Dhariwal, 2018; Kong et al., 2020a; Ping et al., 2020).

To deal with this limitation, several methods have been proposed to reduce the length of the reverse process to $S \ll T$ steps. One class of methods compute continuous noise levels based on discrete diffusion steps and retrain a new model conditioned on these continuous noise levels (Song & Ermon, 2019; Chen et al., 2020; Okamoto et al., 2021; San-Roman et al., 2021). Then, a shorter reverse process can be obtained by carefully choosing a small set (size $S$) of noise levels. However, these methods cannot reuse the pretrained diffusion models, because the state-of-the-art DDPM models are conditioned on discrete diffusion steps (Ho et al., 2020; Dhariwal & Nichol, 2021). It is also unclear the diffusion models conditioned on continuous noise levels can achieve comparable sample quality as the state-of-the-art DDPMs on challenging unconditional image and audio synthesis tasks (Dhariwal & Nichol, 2021; Kong et al., 2020b). Another class of methods directly approximate the original reverse process of DDPM models with shorter ones (of length $S$), which are conditioned on discrete diffusion steps (Song et al., 2020a; Kong et al., 2020b). Although both classes of methods have shown the trade-off between sampling speed and sample quality (i.e., larger $S$ lead to higher sample quality), the fast sampling methods without retraining are more advantageous for fast iteration and de-

ployment, while still keeping high-fidelity synthesis with small number of steps in the reverse process (e.g., $S = 6$ in Kong et al. (2020b)).

In this work, we propose FastDPM, a unified framework of fast sampling methods for diffusion models without retraining. The core idea of FastDPM is to *i)* generalize discrete diffusion steps to continuous diffusion steps, and *ii)* design a bijective mapping between continuous diffusion steps and continuous noise levels. Then, we use this bijection to construct an approximate diffusion process and an approximate reverse process, both of which have length $S \ll T$.

FastDPM includes and generalizes the fast sampling algorithms from denoising diffusion implicit models (DDIM) (Song et al., 2020a) and DiffWave (Kong et al., 2020b). In detail, FastDPM offers two ways to construct the approximate diffusion process: selecting $S$ steps in the original diffusion process, or more flexibly, choosing $S$ variances. FastDPM also offers ways to construct the approximate reverse process: using the stochastic DDPM reverse process (DDPM-rev), or using the implicit (deterministic) DDIM reverse process (DDIM-rev). We can control the amount of stochasticity in the reverse process of FastDPM as in Song et al. (2020a).

FastDPM gives rise to new algorithms with improved sample quality than previous methods when the length of the approximate reverse process $S$ is small. We then extensively evaluate the family of FastDPM methods across image and audio domains. We find the deterministic DDIM-rev significantly outperforms the stochastic DDPM-rev in image generation tasks, but DDPM-rev significantly outperforms DDIM-rev in audio synthesis tasks. Finally, we investigate the performance of different methods by varying the amount of conditional information. We find with different amount of conditional information, we need different amount of stochasticity in the reverse process of FastDPM.

We discuss related work in Appendix A.

## 2. Diffusion Models

Let $d$ be the data dimension. Let $p_{\text{data}}$ be the data distribution and $p_{\text{latent}} = \mathcal{N}(0, I_{d \times d})$ be the latent distribution. Then, the denoising diffusion probabilistic model (DDPM, Sohl-Dickstein et al., 2015; Ho et al., 2020) is a deep generative model consisting two Markov chains called diffusion and reverse processes, respectively. The length of each Markov chain is $T$, which is called the number of diffusion or reverse steps. The *diffusion process* gradually adds Gaussian noise to the data distribution until the noisy data distribution is close to the latent distribution. Formally, the diffusion process from data $x_0 \sim p_{\text{data}}$ to the latent variable $x_T$ is defined as $q(x_1, \cdots, x_T | x_0) = \prod_{t=1}^{T} q(x_t | x_{t-1})$, where each of $q(x_t | x_{t-1}) = \mathcal{N}(x_t; \sqrt{1 - \beta_t} x_{t-1}, \beta_t I)$ for some

small constant $\beta_t > 0$. The hyperparameters $\beta_1, \cdots, \beta_T$ are called the *variance schedule*.

The *reverse process* aims to eliminate the noise added in each diffusion step. Formally, the reverse process from $x_T \sim p_{\text{latent}}$ to $x_0$ is defined as $p_\theta(x_0, \cdots, x_{T-1} | x_T) = \prod_{t=1}^{T} p_\theta(x_{t-1} | x_t)$, where each of $p_\theta(x_{t-1} | x_t)$ is defined as $\mathcal{N}(x_{t-1}; \mu_\theta(x_t, t), \sigma_t^2 I)$; the mean $\mu_\theta(x_t, t)$ is parameterized through a neural network and the variance $\sigma_t$ is time-step dependent constant. Based on the reverse process, the sampling process is to first draw $x_T \sim \mathcal{N}(0, I)$, then draw $x_{t-1} \sim p_\theta(x_{t-1} | x_t)$ for $t = T, T-1, \cdots, 1$, and finally outputs $x_0$.

The training objective of DDPM is based on the variational evidence lower bound (ELBO). Under a certain parameterization introduced by Ho et al. (2020), the objective can be largely simplified. One may first define constants $\alpha_t = 1 - \beta_t$, $\bar{\alpha}_t = \prod_{i=1}^{t} \alpha_i$, $\tilde{\beta}_t = \frac{1 - \bar{\alpha}_{t-1}}{1 - \bar{\alpha}_t} \beta_t$ for $t > 1$ and $\tilde{\beta}_1 = \beta_1$. Then, a noticeable property of diffusion model is

$$q(x_t | x_0) = \mathcal{N}(x_t; \sqrt{\bar{\alpha}_t} x_0, \ (1 - \bar{\alpha}_t)I), \qquad (1)$$

thus one can directly sample $x_t$ given $x_0$ (see Appendix B.1 for derivation). Furthermore, one may parameterize $\mu_\theta(x_t, t) = \frac{1}{\sqrt{\alpha_t}} \left( x_t - \frac{\beta_t}{\sqrt{1 - \bar{\alpha}_t}} \epsilon_\theta(x_t, t) \right)$, where $\epsilon_\theta$ is a neural network taking $x_t$ and the diffusion-step $t$ as inputs. In addition, $\sigma_t$ is simply parameterized as $\tilde{\beta}_t^{\frac{1}{2}}$. Ho et al. (2020) show that minimizing the following unweighted variant of the ELBO leads to higher generation quality:

$$\min_\theta L_{\text{unweighted}}(\theta) = \mathbb{E}_{x_0, \epsilon, t} \|\epsilon - \epsilon_\theta(x_t, \ t)\|_2^2, \qquad (2)$$

where $\epsilon \sim \mathcal{N}(0, I)$, $x_0 \sim q_{\text{data}}$, $t$ is uniformly taken from $1, \cdots, T$, and $x_t = \sqrt{\bar{\alpha}_t} \cdot x_0 + \sqrt{1 - \bar{\alpha}_t} \cdot \epsilon$ from Eq. (1).

## 3. FastDPM: A Unified Framework for Fast Sampling in Diffusion Models

In order to achieve high-fidelity synthesis, the number of diffusion steps $T$ in DDPM is set to be very large so that $q(x_T | x_0)$ is close to $p_{\text{latent}}$. For example, $T = 1000$ in image synthesis (Ho et al., 2020) and $T = 200$ in audio synthesis (Kong et al., 2020b). Then, sampling from DDPM needs running through the network $\epsilon_\theta$ for as many as $T$ times, which can be very slow. In this section, we propose FastDPM, which approximates the pretrained DDPM via much shorter diffusion and reverse processes of length $S \ll T$, thus it can generate a sample by only running the network $S$ times. The core idea of FastDPM is to: *i)* generalize discrete diffusion steps to continuous diffusion steps and, then *ii)* design a bijective mapping between continuous diffusion steps and continuous noise levels, where these noise levels indicate the amount of noise in data. Finally, we use this bijective mapping to construct an approximate diffusion process and an approximate reverse process, respectively.

## 3.1. Bijective mapping between Continuous Diffusion Steps and Noise Levels

In this section, we generalize discrete (integer) diffusion steps to continuous (real-valued) diffusion steps. Then, we introduce a bijective mapping $\mathcal{R}$ and $\mathcal{T} = R^{-1}$ between continuous diffusion steps $t$ and noise levels $r$.

**Define $\mathcal{R}$.** We start with an integer diffusion step $t$. From Eq. (1), one can observe $x_t = \sqrt{\bar{\alpha}_t} \cdot x_0 + \sqrt{1 - \bar{\alpha}_t} \cdot \epsilon$ where $\epsilon \sim \mathcal{N}(0, I)$, thus sampling $x_t$ given $x_0$ is equivalent to adding a Gaussian noise to $x_0$. Based on this observation, we define the noise level at step $t$ as $\mathcal{R}(t) = \sqrt{\bar{\alpha}_t}$, which means $x_t$ is composed of $\mathcal{R}(t)$ fraction of the data $x_0$ and $(1 - \mathcal{R}(t))$ fraction of white noise. For example, $\mathcal{R}(t) = 0$ means no noise and $\mathcal{R}(t) = 1$ means pure white noise. Next, we extend the domain of $\mathcal{R}$ to real values. Assume that the variance schedule $\{\beta_t\}_{t=1}^T$ is linear: $\beta_i = \beta_1 + (i - 1)\Delta\beta$, where $\Delta\beta = \frac{\beta_T - \beta_1}{T - 1}$ (Ho et al., 2020). We further define an auxiliary constant $\hat{\beta} = \frac{1 - \beta_1}{\Delta\beta}$, which is $\gg T$ assuming that $\beta_T \ll 1.0$. [1] Then, we have $\bar{\alpha}_t = (\Delta\beta)^t \Gamma\left(\hat{\beta} + 1\right) / \Gamma\left(\hat{\beta} - t + 1\right)$ (see Appendix B.2). Because the Gamma function $\Gamma$ is well-defined on $(0, \infty)$, it gives rise to a natural extension of $\bar{\alpha}_t$ for continuous diffusion steps $t$. As a result, for $t \in [0, \hat{\beta})$, we define the noise level at $t$ as:

$$\mathcal{R}(t) = (\Delta\beta)^{\frac{t}{2}} \Gamma\left(\hat{\beta} + 1\right)^{\frac{1}{2}} \Gamma\left(\hat{\beta} - t + 1\right)^{-\frac{1}{2}}. \quad (3)$$

**Define $\mathcal{T}$.** For any noise level $r \in (0, 1)$, its corresponding (continuous) diffusion step, $\mathcal{T}(r)$, is defined by inverting $\mathcal{R}$: $\mathcal{T}(r) = \mathcal{R}^{-1}(r)$. Given a noise level $r = \mathcal{R}(t)$, we numerically solve $t = \mathcal{T}(r)$ by applying a binary search based on Eq. (3). We have $\mathcal{T}(r) \in [t, t + 1]$ for $r \in [\sqrt{\bar{\alpha}_{t+1}}, \sqrt{\bar{\alpha}_t}]$, and this provides a good initialization to the binary search algorithm. Experimentally, we find the binary search algorithm converges in no more than 20 iterations.

## 3.2. Approximate the Diffusion Process

Let $\hat{x}_0 \sim p_{\text{data}}$. Given a sequence of noise levels $1 > r_1 > r_2 > \cdots > r_S > 0$, we aim to construct each step in the approximate diffusion process as $\hat{x}_s \sim \mathcal{N}(\hat{x}_s; r_s \hat{x}_0, (1 - r_s^2)I)$. To achieve this goal, we define $\gamma_s = r_s^2 / r_{s-1}^2$, compute the corresponding variances as $\eta_s = 1 - \gamma_s = 1 - r_s^2 / r_{s-1}^2$, and then define the transition probability in the approximate diffusion process as

$$q(\hat{x}_s | \hat{x}_{s-1}) = \mathcal{N}(\hat{x}_s; \sqrt{1 - \eta_s}\hat{x}_{s-1}, \eta_s I)$$
$$= \mathcal{N}\left(\hat{x}_s; \frac{r_s}{r_{s-1}}\hat{x}_{s-1}, \left(1 - \frac{r_s^2}{r_{s-1}^2}\right)I\right). \quad (4)$$

One can see this by rewriting Eq. (1): $\eta_s$ corresponds to $\beta_t = 1 - \alpha_t$, $\gamma_s$ corresponds to $\alpha_t$, and $r_s$ corresponds to

$\sqrt{\bar{\alpha}_t}$. We then propose the following two ways to schedule the noise levels $\{r_s\}_{s=1}^S$.

**Noise levels from variances (VAR).** We start from the variance schedule $\{\eta_s\}_{s=1}^S$. Next, we compute $\gamma_s = 1 - \eta_s$ and $\bar{\gamma}_s = \prod_{i=1}^s \gamma_i$. The noise level at step $s$ is $r_s = \sqrt{\bar{\gamma}_s}$.

**Noise levels from steps (STEP).** We start from a subset of diffusion steps $\{\tau_s\}_{s-1}^S$ in $\{1, \cdots, T\}$. Then, the noise level at step $s$ is $r_s = \mathcal{R}(\tau_s) = \sqrt{\bar{\alpha}_{\tau_s}}$.

When $\eta_s = 1 - \bar{\alpha}_{\tau_s} / \bar{\alpha}_{\tau_{s-1}}$, we have $\bar{\gamma}_s = \bar{\alpha}_{\tau_s}$. Therefore, noise levels from steps can be regarded as a special case of noise levels from variances.

## 3.3. Approximate the Reverse Process

Given the same sequence of noise levels in Section 3.2, we aim to approximate the reverse process in the original DDPM. To achieve this goal, we regard the model $\epsilon_\theta$ as being trained on variances $\{\eta_s\}_{s=1}^S$ instead of the original $\{\beta_t\}_{t=1}^T$. Then, the transition probability in the approximate reverse process is

$$p_\theta(\hat{x}_{s-1} | \hat{x}_s) = \mathcal{N}\left(\hat{x}_{s-1}; \hat{\mu}(\hat{x}_s, s), \tilde{\eta}_s I\right), \quad (5)$$

where $\hat{\mu}(\hat{x}_s, s) = \frac{1}{\sqrt{\gamma_s}}\left(\hat{x}_s - \frac{\eta_s}{\sqrt{1 - \bar{\gamma}_s}}\epsilon_\theta(\hat{x}_s, \mathcal{T}(r_s))\right)$, $\tilde{\eta}_s = \frac{1 - \bar{\gamma}_{s-1}}{1 - \bar{\gamma}_s}\eta_s$ for $s > 1$ and $\tilde{\eta}_1 = \eta_1$. $\tilde{\eta}_s$ corresponds to the $\tilde{\beta}_t = \sigma_t^2$ term. There are two ways to sample from the approximate reverse process in Eq. (5). Let every $\hat{\epsilon}_s$ be i.i.d. standard Gaussians for $1 \leq s \leq S$.

**DDPM reverse process (DDPM-rev).** The sampling procedure based on the DDPM reverse process is based on Eq. (5): that is, to first sample $\hat{x}_S \sim p_{\text{latent}}$ and then sample $\hat{x}_{s-1} = \hat{\mu}(\hat{x}_s, s) + \sqrt{\tilde{\eta}_s}\hat{\epsilon}_s$.

**DDIM reverse process (DDIM-rev).** Let $\kappa \in [0, 1]$ be a hyperparameter. [2] Then, the sampling procedure based on DDIM (Song et al., 2020a) is to first sample $\hat{x}_S \sim p_{\text{latent}}$ and then sample $\hat{x}_{s-1} = \sqrt{\bar{\gamma}_{s-1}}\left(\frac{\hat{x}_s - \sqrt{1 - \bar{\gamma}_s}\epsilon_\theta(\hat{x}_s, \mathcal{T}(r_s))}{\sqrt{\bar{\gamma}_s}}\right) + \sqrt{1 - \bar{\gamma}_{s-1} - \kappa^2 \tilde{\eta}_s}\epsilon_\theta(\hat{x}_s, \mathcal{T}(r_s)) + \kappa\sqrt{\tilde{\eta}_s}\hat{\epsilon}_s$. When $\kappa = 1$, it is exactly DDPM-rev (see Appendix B.3 for derivation).

## 3.4. Connections with Previous Methods

The DDIM (Song et al., 2020a) method is equivalent to STEP + DDIM-rev in FastDPM. The fast sampling algorithm by DiffWave (Kong et al., 2020b) is related to VAR + DDPM-rev in FastDPM. Compared with DiffWave, FastDPM offers an automatic way to select variances in different settings and a more natural way to compute noise levels.

---

[1] E.g., $\beta_T = 0.02$ in Ho et al. (2020); Kong et al. (2020b).

[2] $\kappa$ is $\eta$ in Song et al. (2020a).

# 4. Experiments

In this section, we aim to answer the following two questions for FastDPM. (1) Which approximate diffusion process, VAR or STEP, is better? And (2) which approximate reverse process, DDPM-rev or DDIM-rev, is better? We investigate these questions by conducting extensive experiments in both image and audio domains.

We conduct unconditional image generation experiments on CIFAR-10 (Krizhevsky et al., 2009), CelebA (Liu et al., 2015), and LSUN-bedroom (Yu et al., 2015), unconditional and class-conditional audio synthesis experiments on the Speech Commands 0-9 (SC09) dataset (Warden, 2018), and neural vocoding experiments (audio synthesis conditioned on mel spectrogram) on the LJSpeech dataset (Ito, 2017).

We use pretrained models in all experiments (Ho et al., 2020; Esser, 2020; Song et al., 2020a; Kong et al., 2020b). We use Fréchet Inception Distance (FID) (Heusel et al., 2017; Lang, 2020), Inception Score (IS) (Salimans et al., 2016), and the crowdMOS tookit (Ribeiro et al., 2011) to evaluate generated samples. Details of experimental setup can be found in Appendix C. Results can be found in Appendix D. Generated samples can be found in Appendix E and the demo website. [3]

## 4.1. Observations and Insights

We have the following observations and insights according to the above experimental results.

**VAR marginally outperforms STEP for small $S$.** In the above experiments, the two approximate diffusion processes (STEP and VAR) generally match performances of each other. On CIFAR-10, VAR outperforms STEP when $S = 10$, and STEP slightly outperforms VAR when $S \geq 20$. On CelebA, VAR slightly outperforms STEP when $S \leq 20$, and they have similar results when $S \geq 50$. On LSUN-bedroom, VAR slightly outperforms STEP when $S \leq 50$, and STEP slightly outperforms VAR when $S = 100$. On SC09, VAR slightly outperforms STEP in most cases. On LJSpeech, VAR slightly outperforms STEP when $S = 5$. Based on these results, we conclude that VAR marginally outperforms STEP for small $S$.

**Different reverse processes dominate in different domains.** In the above experiments, the difference between DDPM and DDIM reverse processes is very clear. In image generation tasks, DDIM-rev significantly outperforms DDPM-rev except for the $S = 100$ case in the LSUN-bedroom experiment. When we *reduce* $\kappa$ from 1.0 to 0.0 (see Table 1), the quality of generated samples consistently improves. In contrast, in audio synthesis tasks, DDPM-rev

---

[3]Demo website: https://fastdpm.github.io. Code: https://github.com/FengNiMa/FastDPM_pytorch

significantly outperforms DDIM-rev. When we *increase* $\kappa$ from 0.0 to 1.0 (see Table 4), the quality of generated samples consistently improves. This can also be observed from Figure 8: DDIM produces very noisy utterances while DDPM produces very clean utterances.

The results indicate that in the image domain, DDIM-rev produces better quality whereas in the audio domain, DDPM-rev produces better quality. We speculate the reason behind the difference is that in the audio domain, waveforms naturally exhibit significant amount of stochasticity. The DDPM reverse process offers much stochasticity because at each reverse step $s$, $\hat{x}_{s-1}$ is sampled from a Gaussian distribution. However, the DDIM reverse process ($\kappa = 0.0$) is a deterministic mapping from latents to data, so it leads to degrade quality in the audio domain. This hypothesis is also aligned with previous result that the flow-based model with deterministic mapping was unable to generate intelligible speech unconditionally on SC09 (Ping, 2021).

**The amount of conditional information affects the choice of reverse processes.** In audio synthesis experiments, we find the amount of conditional information affects the generation quality of FastDPM with different reverse processes. In the unconditional generation experiment on SC09, DDPM-rev (i.e. $\kappa = 1.0$) has the best results. When there is slightly more conditional information in the class-conditional generation experiment on SC09, DDIM-rev with $\kappa = 0.5$ has the best results and slightly outperforms DDPM-rev. In both experiments DDIM-rev with $\kappa = 0.0$ has much worse results. When there is much more conditional information (mel spectrogram) in the neural vocoding experiments on LJSpeech, DDPM-rev is still better than DDIM-rev, but the difference between these two methods is reduced. We speculate that adding conditional information reduces the amount of stochasticity required. When there is no conditional information, we need maximum stochasticity ($\kappa = 1.0$); with weak class information, we need moderate stochasticity ($\kappa = 0.5$); and with strong mel-spectrogram information, even having no stochasticity ($\kappa = 0.0$) is able to generate reasonable samples.

# 5. Conclusion

Diffusion models are a class of powerful deep generative models that produce superior quality samples on various generation tasks. In this paper, we introduce FastDPM, a unified framework for fast sampling in diffusion models without retraining. FastDPM generalizes prior methods and provides more flexibility. We extensively evaluate and analyze FastDPM in image and audio generation tasks. One limitation of FastDPM is that when $S$ is small, there is still quality degradation compared to the original DDPM. We plan to study algorithms offering higher quality for extremely small $S$ in future.

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

# A. Related Work

Diffusion models are a class of powerful deep generative models (Sohl-Dickstein et al., 2015; Ho et al., 2020; Goyal et al., 2017), which have received a lot of attention recently. These models have been applied to various domains, including image generation (Ho et al., 2020; Dhariwal & Nichol, 2021), audio synthesis (Kong et al., 2020b; Chen et al., 2020; Okamoto et al., 2021), image or audio super-resolution (Li et al., 2021; Lee & Han, 2021), text-to-speech (Jeong et al., 2021; Popov et al., 2021), music synthesis (Liu et al., 2021; Mittal et al., 2021), 3-D point cloud generation (Luo & Hu, 2021; Zhou et al., 2021), and language models (Hoogeboom et al., 2021). Diffusion models are connected with scored-based models (Song & Ermon, 2019; 2020; Song et al., 2020b), and there have been a series of research extending and improving diffusion models (Song et al., 2020b; Gao et al., 2020; Dhariwal & Nichol, 2021; San-Roman et al., 2021; Meng et al., 2021).

There are two families of methods aiming for accelerating diffusion models at synthesis, which reduce the length of the reverse process from $T$ to a much smaller $S$. One family of methods tackle this problem at training. They retrain the network conditioned on continuous noise levels instead of discrete diffusion steps (Song & Ermon, 2019; Chen et al., 2020; Okamoto et al., 2021; San-Roman et al., 2021). Assuming that the corresponding network is able to predict added noise at any noise level, we can carefully choose only $S \ll T$ noise levels and construct a short reverse process just based on them. San-Roman et al. (2021) present a learning scheme that can step-by-step adjust those noise level parameters, for any given number of steps $S$. Another family of methods aim to directly approximate the original reverse process within the pretrained DDPM conditioned on discrete steps. In other words, no retraining is needed. Song et al. (2020a) introduce denoising diffusion implicit models (DDIM), which contain non-Markovian processes that lead to an equivalent training objective as DDPM. These non-Markovian processes naturally permit "jumping steps", or formally, using a subset of steps to form a short reverse process. However, compared to using continuous noise levels, selecting discrete steps offers less flexibility. Kong et al. (2020b) introduce a fast sampling algorithm by interpolating steps according to corresponding noise levels. This can be seen as an attempt to map continuous noise levels to discrete diffusion steps. However, it lacks both theoretical justification for the interpolation and extensive empirical studies.

In this paper, we propose FastDPM, a method that approximates the original DDPM model. FastDPM constructs a bijective mapping between (continuous) diffusion steps and continuous noise levels. This allows us to take advantage of the flexibility of using these continuous noise levels. FastDPM generalizes Kong et al. (2020b) by using Gamma functions to compute noise levels, which naturally extends from discrete domain to continuous domain. FastDPM generalizes Song et al. (2020a) by providing a special set of noise levels that exactly correspond to integer steps.

# B. Derivations

## B.1. Derivation of $q(x_t|x_0)$

According to the definition of diffusion process, we have

$$x_t = \sqrt{\alpha_t} x_{t-1} + \sqrt{\beta_t} \epsilon_t, \tag{6}$$

where each $\epsilon_t$ is an i.i.d. standard Gaussian. Then, by recursion, we have

$$
\begin{aligned}
x_t &= \sqrt{\alpha_t \alpha_{t-1}} x_{t-2} + \sqrt{\alpha_t \beta_{t-1}} \epsilon_{t-1} + \sqrt{\beta_t} \epsilon_t \\
&= \sqrt{\alpha_t \alpha_{t-1} \alpha_{t-1}} x_{t-3} + \sqrt{\alpha_t \alpha_{t-1} \beta_{t-2}} \epsilon_{t-2} + \sqrt{\alpha_t \beta_{t-1}} \epsilon_{t-1} + \sqrt{\beta_t} \epsilon_t \\
&\vdots \\
&= \sqrt{\bar{\alpha_t}} x_0 + \sqrt{\alpha_t \alpha_{t-1} \cdots \alpha_2 \beta_1} \epsilon_1 + \cdots + \sqrt{\alpha_t \beta_{t-1}} \epsilon_{t-1} + \sqrt{\beta_t} \epsilon_t.
\end{aligned}
\tag{7}
$$

As a result, $q(x_t|x_0)$ is still Gaussian. Its mean vector is $\sqrt{\bar{\alpha_t}} x_0$, and its covariance matrix is $(\alpha_t \alpha_{t-1} \cdots \alpha_2 \beta_1 + \cdots + \alpha_t \beta_{t-1} + \beta_t) I = (1 - \bar{\alpha}_t) I$. Formally, we have

$$q(x_t|x_0) = \mathcal{N}(x_t; \sqrt{\bar{\alpha_t}} x_0, (1 - \bar{\alpha}_t) I). \tag{8}$$

## B.2. Derivation of $\bar{\alpha}_t$

$$
\begin{aligned}
\bar{\alpha}_t &= \prod_{i=1}^{t} (1 - \beta_i) \\
&= \prod_{i=1}^{t} (1 - \beta_1 - (i-1)\Delta\beta) \\
&= (\Delta\beta)^t \prod_{i=0}^{t-1} \left( \hat{\beta} - i \right) \\
&= (\Delta\beta)^t \Gamma\left( \hat{\beta} + 1 \right) \Gamma\left( \hat{\beta} - t + 1 \right)^{-1}.
\end{aligned}
\tag{9}
$$

## B.3. Derivation of DDIM ($\kappa = 1$)

When $\kappa = 1$, the coefficient of the $\epsilon_\theta$ term in the DDIM reverse process is

$$
\begin{aligned}
-\frac{\sqrt{1 - \bar{\gamma}_s}}{\sqrt{\gamma_s}} + \sqrt{1 - \bar{\gamma}_{s-1} - \frac{1 - \bar{\gamma}_{s-1}}{1 - \bar{\gamma}_s} \eta_s} &= -\frac{1 - \bar{\gamma}_s}{\sqrt{\gamma_s(1 - \bar{\gamma}_s)}} + \frac{\sqrt{(\gamma_s - \bar{\gamma}_s)(1 - \bar{\gamma}_s - \eta_s)}}{\sqrt{\gamma_s(1 - \bar{\gamma}_s)}} \\
&= -\frac{1 - \bar{\gamma}_s}{\sqrt{\gamma_s(1 - \bar{\gamma}_s)}} + \frac{\gamma_s - \bar{\gamma}_s}{\sqrt{\gamma_s(1 - \bar{\gamma}_s)}} \\
&= -\frac{\eta_s}{\sqrt{\gamma_s(1 - \bar{\gamma}_s)}}.
\end{aligned}
\tag{10}
$$

# C. Detailed Experimental Setup

**Image datasets.** We conduct unconditional image generation experiments on three datasets: CIFAR-10 (50k object images of resolution $32 \times 32$ (Krizhevsky et al., 2009)), CelebA ($\sim$163k face images of resolution $64 \times 64$ (Liu et al., 2015)), and LSUN-bedroom ($\sim$3M bedroom images of resolution $256 \times 256$ (Yu et al., 2015)).

**Audio datasets.** We conduct unconditional and class-conditional audio synthesis experiments on the Speech Commands 0-9 (SC09) dataset, the spoken digit subset of the full Speech Commands dataset (Warden, 2018). SC09 contains $\sim$31k one-second long utterances of ten classes (0 through 9) with a sampling rate of 16kHz. We conduct neural vocoding experiments (audio synthesis conditioned on mel spectrogram) on the LJSpeech dataset (Ito, 2017). It contains $\sim$24 hours of audio ($\sim$13k utterances from a female speaker) recorded in home environment with a sampling rate of 22.05kHz.

**Models.** In all experiments, we use pretrained checkpoints in prior works. In detail, the pretrained models for CIFAR-10 and LSUN-bedroom are taken from DDPM (Ho et al., 2020; Esser, 2020), the pretrained model for CelebA is taken from DDIM (Song et al., 2020a). In these models, $T$ is 1000. The pretrained models for SC09 and LJSpeech are taken from DiffWave (Kong et al., 2020b). In these models, $T$ is 200. In all models, $\beta_1 = 10^{-4}$, $\beta_T = 2 \times 10^{-2}$, and all $\beta_t$'s are linearly interpolated between $\beta_1$ and $\beta_T$.

**Noise level schedules.** For each of the approximate diffusion process in Section 3.2, we examine two schedules: linear and quadratic. For noise levels $\{\eta_s\}_{s=1}^{S}$ from variances, the two schedules are:

- Linear (VAR): $\eta_s = (1 + cs)\, \eta_0$.

- Quadratic (VAR): $\eta_s = (1 + cs)^2\, \eta_0$.

We let $\eta_0 = \beta_0$ and the constant $c$ satisfy $\prod_{s=1}^{S}(1 - \eta_s) = \bar{\alpha}_T$. The noise level at step $s$ is $r_s = \sqrt{\gamma_s}$.

For noise levels $\{\eta_s\}_{s=1}^{S}$ from steps, they are computed from selected steps $\{\tau_s\}_{s=1}^{S}$ among $\{1, \cdots, T\}$ (Song et al., 2020a). The two schedules are:

- Linear (STEP): $\tau_s = \lfloor cs \rfloor$, where $c = \frac{T}{S}$.

- Quadratic (STEP): $\tau_s = \lfloor cs^2 \rfloor$, where $c = \frac{4}{5} \cdot \frac{T}{S^2}$.

Then, the noise level at step $s$ is $r_s = \mathcal{R}(\tau_s) = \sqrt{\bar{\alpha}_{\tau_s}}$.

In image generation experiments, we follow the same noise level schedules as in Song et al. (2020a): quadratic schedules for CIFAR-10 and linear schedules for CelebA and LSUN-bedroom. We use linear schedules in SC09 experiments and quadratic schedules in LJSpeech experiments; we find these schedules have better quality.

**Evaluations.** In all unconditional generation experiments, we use the Fréchet Inception Distance (FID) (Heusel et al., 2017; Lang, 2020) to evaluate generated samples. For the training set $X_t$ and the set of generated samples $X_g$, the FID between these two sets is defined as

$$\text{FID} = \|\mu_t - \mu_g\|^2 + \text{tr}\left(\Sigma_t + \Sigma_g - 2\sqrt{\Sigma_t \Sigma_g}\right), \tag{11}$$

where $\mu_t, \mu_g$ and $\Sigma_t, \Sigma_g$ are the means and covariances of $X_t, X_g$ after a feature transformation. In each image generation experiment, $X_g$ is 50K generated images. The transformed feature is the 2048-dimensional vector output of the last layer of Inception-V3 (Szegedy et al., 2015). In each audio synthesis experiment, $X_g$ is 5K generated utterances. The transformed feature is the 1024-dimensional vector output of the last layer of a ResNeXT classifier (Xu & Tuguldur, 2017), which achieves 99.06% accuracy on the training set and 98.76% accuracy on the test set. The FID is the smaller the better.

In the class-conditional generation experiment on SC09, we evaluate with accuracy and the Inception Score (IS). [4] The accuracy is computed by matching the predictions of the ResNeXT classifier and the pre-specified labels in the dataset. The IS of generated samples $X_g$ is defined as

$$\text{IS} = \exp\left(\mathbb{E}_{x \sim X_g} \text{KL}(p(x) \| \mathbb{E}_{x' \sim X_g} p(x'))\right), \tag{12}$$

where $p(x)$ is the logit vector of the ResNeXT classifier. The IS and accuracy are the larger the better.

---

[4]Note that FID is not an appropriate metric for conditional generation.

In the neural vocoding experiment on LJSpeech, we evaluate the speech quality with the crowdMOS tookit (Ribeiro et al., 2011), where the test utterances from all models were presented to Mechanical Turk workers. We report the 5-scale Mean Opinion Scores (MOS), and it is the larger the better.

## D. Evaluation Results in Experiments

We report image generation results under different approximate diffusion processes, approximate reverse processes and $S$, the length of FastDPM. Evaluation results on CIFAR-10, CelebA, and LSUN-bedroom measured in FID are shown in Table 1, Table 2, and Table 3, respectively.

*Table 1.* CIFAR-10 image generation measured in FID. STEP means noise levels from steps and VAR means noise levels from variances. Both use quadratic schedules. $S$ is the length of FastDPM. The standard DDPM ($T = 1000$) has FID $= 3.03$.

| Approx. Diffusion | Approx. Reverse | FID ($\downarrow$) | | | |
|---|---|---|---|---|---|
| | | $S = 10$ | $S = 20$ | $S = 50$ | $S = 100$ |
| STEP | DDIM-rev ($\kappa = 0.0$) | 11.01 | **5.05** | **3.20** | **2.86** |
| VAR | DDIM-rev ($\kappa = 0.0$) | **9.90** | 5.22 | 3.41 | 3.01 |
| STEP | DDIM-rev ($\kappa = 0.2$) | 11.32 | 5.16 | 3.27 | 2.87 |
| VAR | DDIM-rev ($\kappa = 0.2$) | 10.18 | 5.32 | 3.50 | 3.04 |
| STEP | DDIM-rev ($\kappa = 0.5$) | 13.53 | 6.14 | 3.61 | 3.05 |
| VAR | DDIM-rev ($\kappa = 0.5$) | 12.22 | 6.55 | 3.86 | 3.15 |
| STEP | DDPM-rev | 36.70 | 14.82 | 5.79 | 4.03 |
| VAR | DDPM-rev | 29.43 | 15.27 | 6.74 | 4.58 |

*Table 2.* CelebA image generation measured in FID. STEP means noise levels from steps and VAR means noise levels from variances. Both use linear schedules. $S$ is the length of FastDPM. The standard DDPM ($T = 1000$) has FID $= 7.00$.

| Approx. Diffusion | Approx. Reverse | FID ($\downarrow$) | | | |
|---|---|---|---|---|---|
| | | $S = 10$ | $S = 20$ | $S = 50$ | $S = 100$ |
| STEP | DDIM-rev ($\kappa = 0.0$) | 15.72 | 10.77 | **8.31** | **7.85** |
| VAR | DDIM-rev ($\kappa = 0.0$) | **15.31** | **10.69** | 8.41 | 7.95 |
| STEP | DDPM-rev | 29.52 | 19.38 | 12.83 | 10.35 |
| VAR | DDPM-rev | 28.98 | 18.89 | 12.83 | 10.39 |

*Table 3.* LSUN-bedroom image generation measured in FID. STEP means noise levels from steps and VAR means noise levels from variances. Both use linear schedules. $S$ is the length of FastDPM.

| Approx. Diffusion | Approx. Reverse | FID ($\downarrow$) | | | |
|---|---|---|---|---|---|
| | | $S = 10$ | $S = 20$ | $S = 50$ | $S = 100$ |
| STEP | DDIM-rev ($\kappa = 0.0$) | **19.07** | 9.95 | 8.43 | 9.94 |
| VAR | DDIM-rev ($\kappa = 0.0$) | 19.98 | **9.86** | **8.37** | 10.27 |
| STEP | DDPM-rev | 42.69 | 20.97 | 10.24 | **7.98** |
| VAR | DDPM-rev | 41.00 | 20.12 | 10.12 | 8.13 |

We report audio synthesis results under different approximate diffusion processes, approximate reverse processes and $S$, the length of FastDPM. Evaluation results of unconditional generation on SC09 measured in FID and IS are shown in Table 4. Evaluation results of class-conditional generation on SC09 measured in accuracy and IS are shown in Table 5. Evaluation results of neural vocoding on LJSpeech measured in MOS are shown in Table 6.

*Table 4.* SC09 unconditional audio synthesis measured in FID and IS. STEP means noise levels from steps and VAR means noise levels from variances. Both use linear schedules. $S$ is the length of FastDPM. The original DiffWave ($T = 200$) has FID $= 1.29$ and IS$= 5.30$.

| Approx. Diffusion | Approx. Reverse | FID ($\downarrow$) | | | IS ($\uparrow$) | | |
|---|---|---|---|---|---|---|---|
| | | $S = 10$ | $S = 20$ | $S = 50$ | $S = 10$ | $S = 20$ | $S = 50$ |
| STEP | DDIM-rev ($\kappa = 0.0$) | 4.72 | 5.31 | 5.54 | 2.46 | 2.27 | 2.23 |
| VAR | DDIM-rev ($\kappa = 0.0$) | 4.74 | 4.88 | 5.58 | 2.49 | 2.42 | 2.21 |
| STEP | DDIM-rev ($\kappa = 0.5$) | 2.60 | 2.52 | 2.46 | 3.94 | 4.17 | 4.19 |
| VAR | DDIM-rev ($\kappa = 0.5$) | 2.67 | 2.49 | 2.47 | 3.94 | 4.20 | 4.20 |
| STEP | DDPM-rev | 1.75 | 1.40 | **1.33** | 4.03 | 4.57 | 5.16 |
| VAR | DDPM-rev | **1.69** | **1.38** | 1.34 | **4.06** | **4.63** | **5.18** |

*Table 5.* SC09 class-conditional audio synthesis. The results are measured by accuracy and IS. STEP means noise levels from steps and VAR means noise levels from variances. Both use linear schedules. $S$ is the length of FastDPM. The DiffWave ($T = 200$) has accuracy $= 91.2\%$ and IS $= 6.63$.

| Approx. Diffusion | Approx. Reverse | Accuracy ($\uparrow$) | | | IS ($\uparrow$) | | |
|---|---|---|---|---|---|---|---|
| | | $S = 10$ | $S = 20$ | $S = 50$ | $S = 10$ | $S = 20$ | $S = 50$ |
| STEP | DDIM-rev ($\kappa = 0.0$) | 66.5% | 68.3% | 66.1% | 3.21 | 3.18 | 2.87 |
| VAR | DDIM-rev ($\kappa = 0.0$) | 66.6% | 68.5% | 66.1% | 3.26 | 3.22 | 2.88 |
| STEP | DDIM-rev ($\kappa = 0.5$) | 85.8% | **88.4%** | 87.8% | **5.79** | 6.23 | 6.00 |
| VAR | DDIM-rev ($\kappa = 0.5$) | **86.0%** | 88.2% | **88.0%** | 5.74 | **6.24** | **6.03** |
| STEP | DDPM-rev | 79.9% | 82.7% | 86.8% | 4.71 | 5.10 | 5.83 |
| VAR | DDPM-rev | 81.0% | 82.8% | 87.0% | 4.93 | 5.16 | 5.86 |

*Table 6.* LJSpeech audio synthesis conditioned on mel spectrogram measured. The results are measured by 5-scale MOS with 95% confidence intervals. STEP means noise levels from steps and VAR means noise levels from variances. Both use quadratic schedules. $S$ is the length of FastDPM.

| Approx. Diffusion | Approx. Reverse | $S$ | MOS ($\uparrow$) |
|---|---|---|---|
| STEP | DDIM-rev ($\kappa = 0.0$) | 5 | $3.72 \pm 0.11$ |
| VAR | DDIM-rev ($\kappa = 0.0$) | 5 | $3.75 \pm 0.10$ |
| STEP | DDPM-rev | 5 | $4.28 \pm 0.08$ |
| VAR | DDPM-rev | 5 | $\mathbf{4.31 \pm 0.07}$ |
| DiffWave ($T = 200$) | | 200 | $4.42 \pm 0.10$ |
| Ground truth | | – | $4.51 \pm 0.07$ |

# E. Generated Samples in Experiments

In this section, we display generated samples of FastDPM, including image samples and mel-spectrogram of audio samples.

## E.1. Unconditional Generation on CIFAR-10

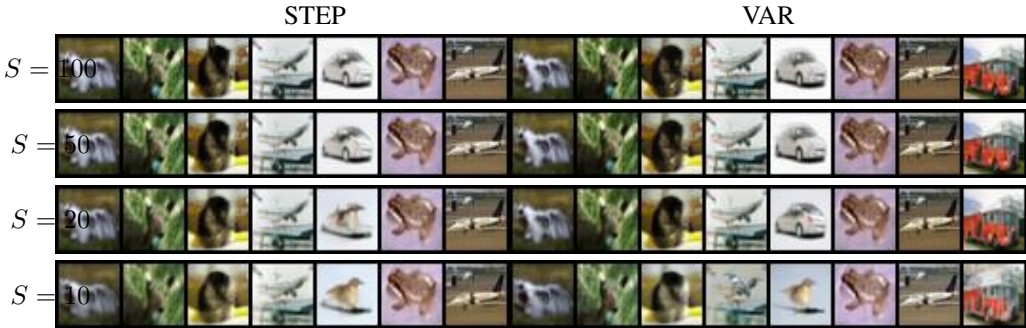

*Figure 1.* Comparison of generated samples of FastDPM on CIFAR-10 among different $S$ and approximate diffusion processes. The approximate reverse process is DDIM-rev ($\kappa = 0.0$).

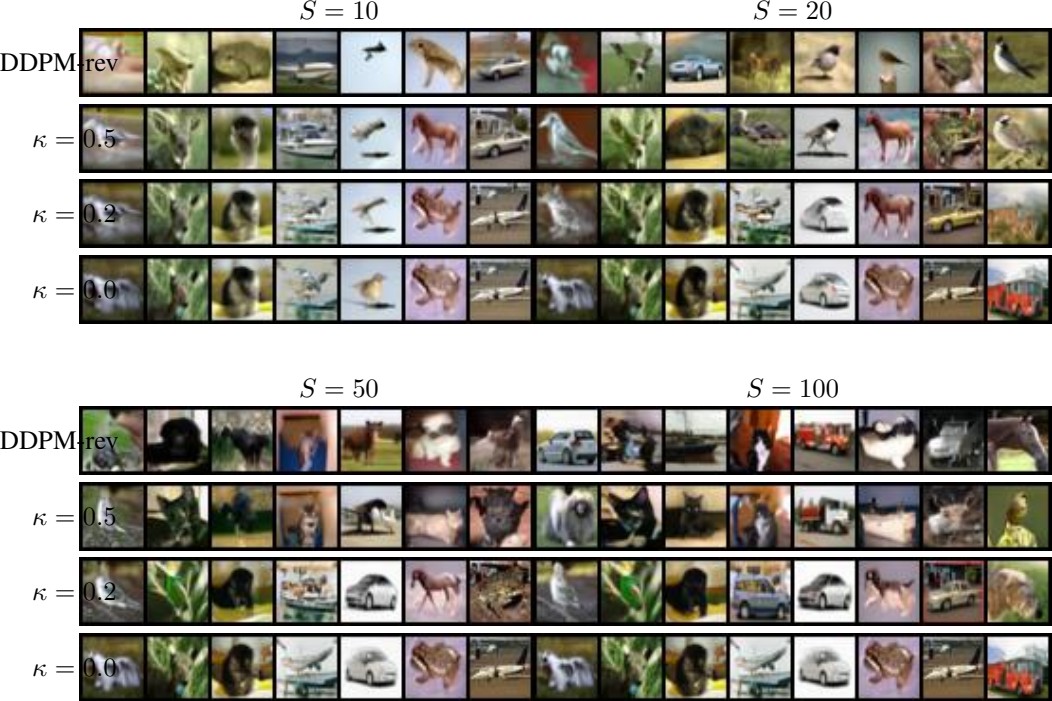

*Figure 2.* Comparison of generated samples of FastDPM on CIFAR-10 among different $S$ and approximate reverse processes. The approximate diffusion process is VAR.

### E.2. Unconditional Generation on CelebA

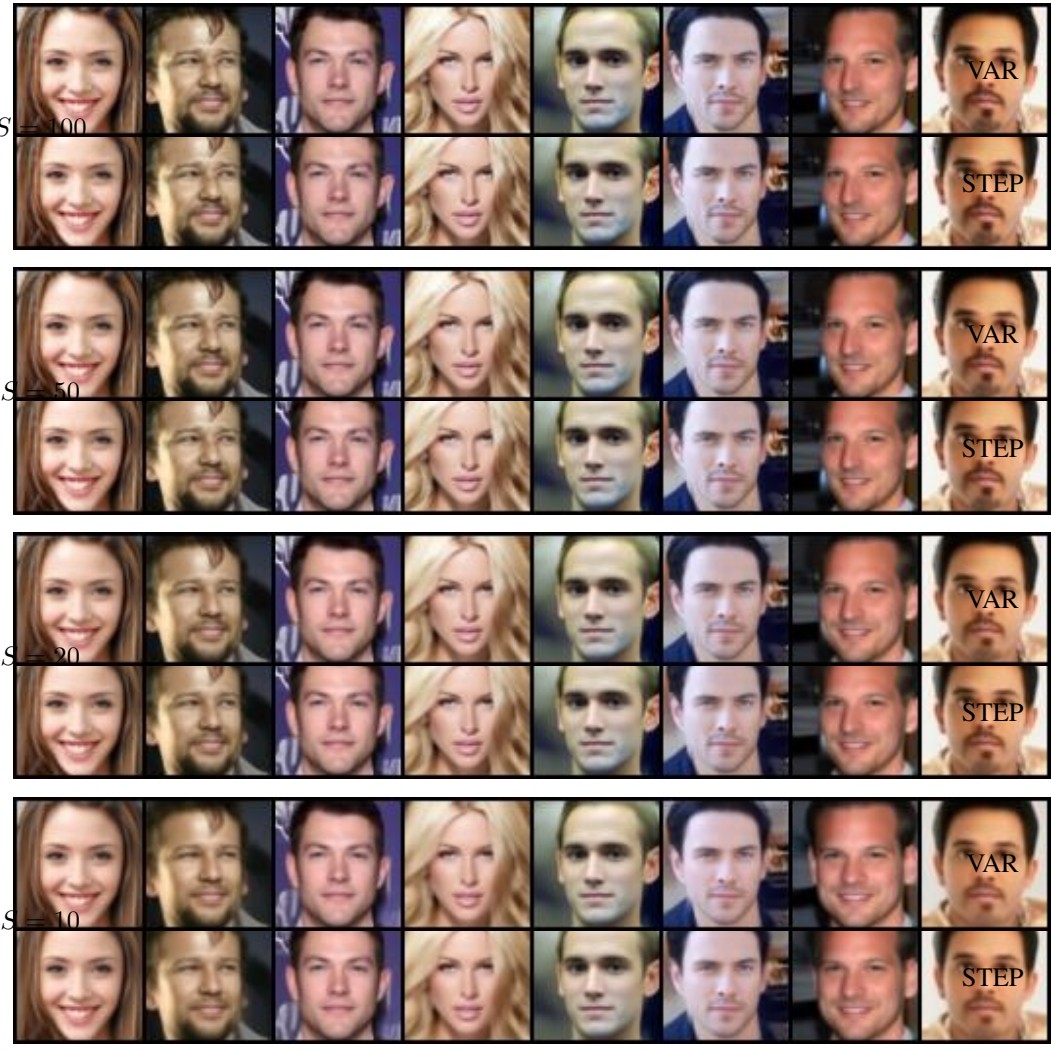

*Figure 3.* Comparison of generated samples of FastDPM on CelebA among different $S$ and approximate diffusion processes. The approximate reverse process is DDIM-rev ($\kappa = 0.0$).

### E.3. Unconditional Generation on LSUN-bedroom

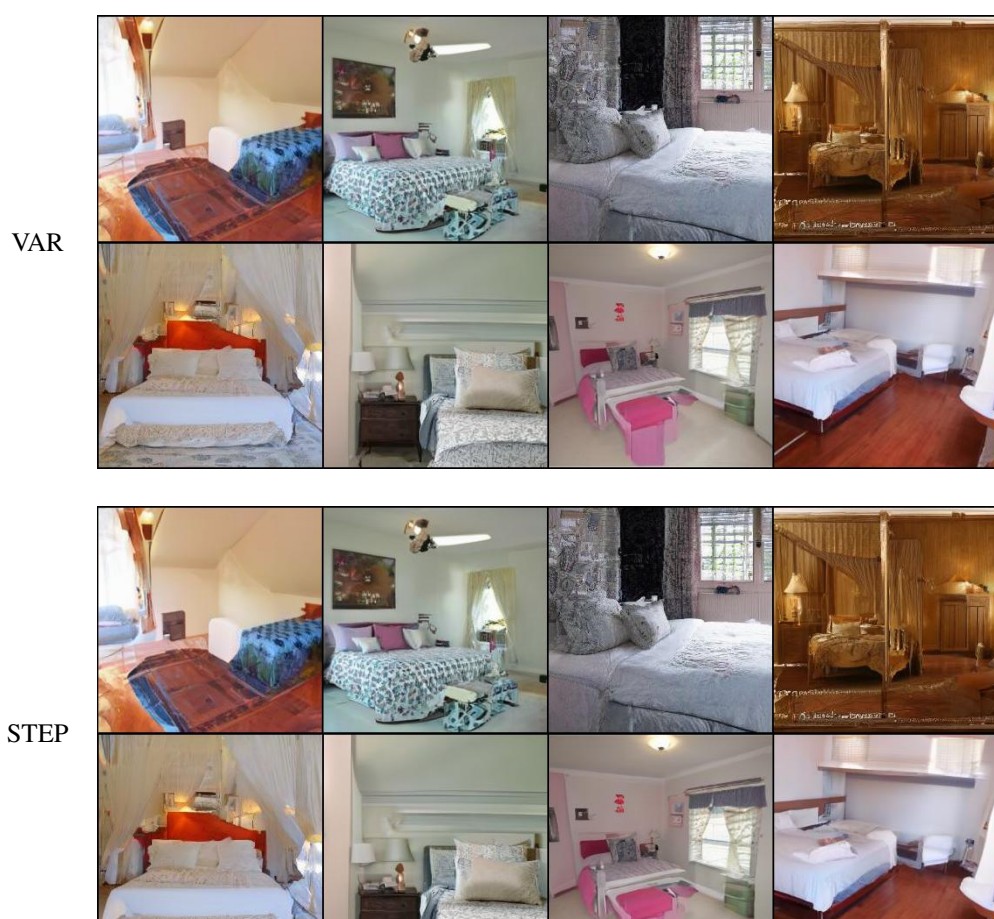

*Figure 4.* Comparison of generated samples of FastDPM on LSUN bedroom among different approximate diffusion processes. The approximate reverse process is DDIM-rev ($\kappa = 0.0$) and $S = 100$.

VAR

STEP

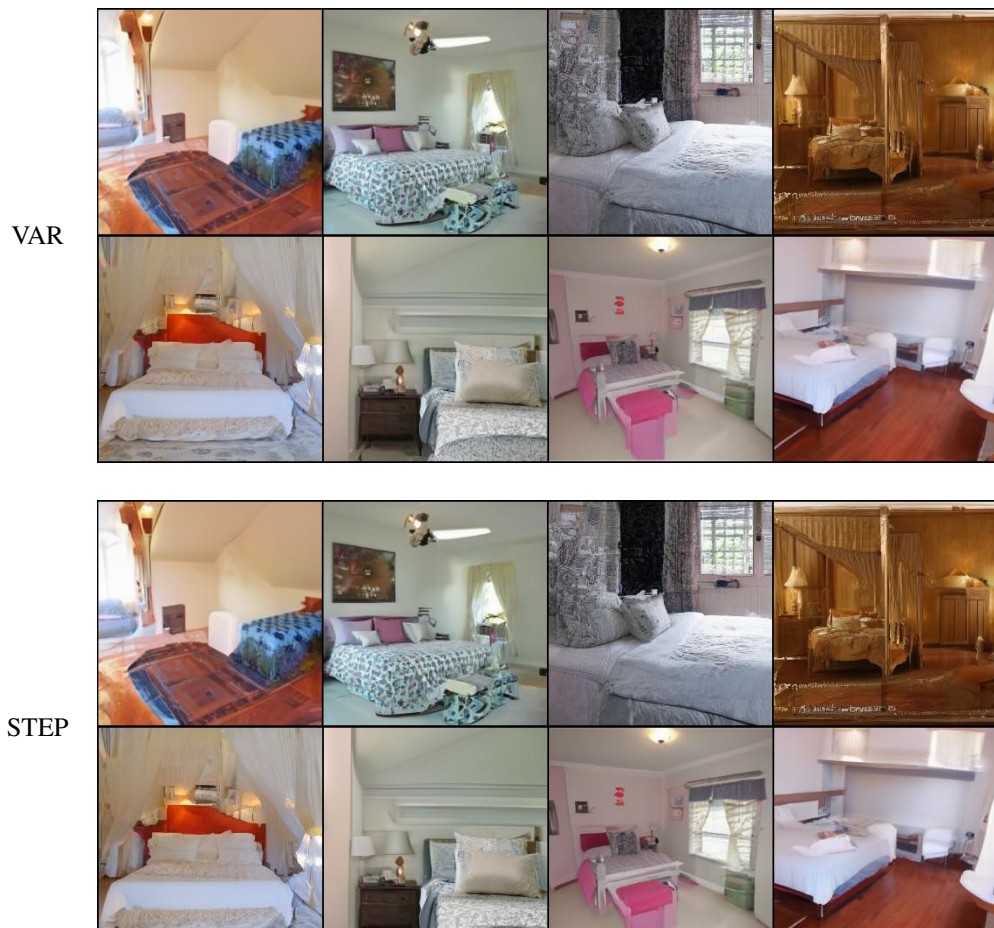

*Figure 5.* Comparison of generated samples of FastDPM on LSUN bedroom among different approximate diffusion processes. The approximate reverse process is DDIM-rev ($\kappa = 0.0$) and $S = 50$.

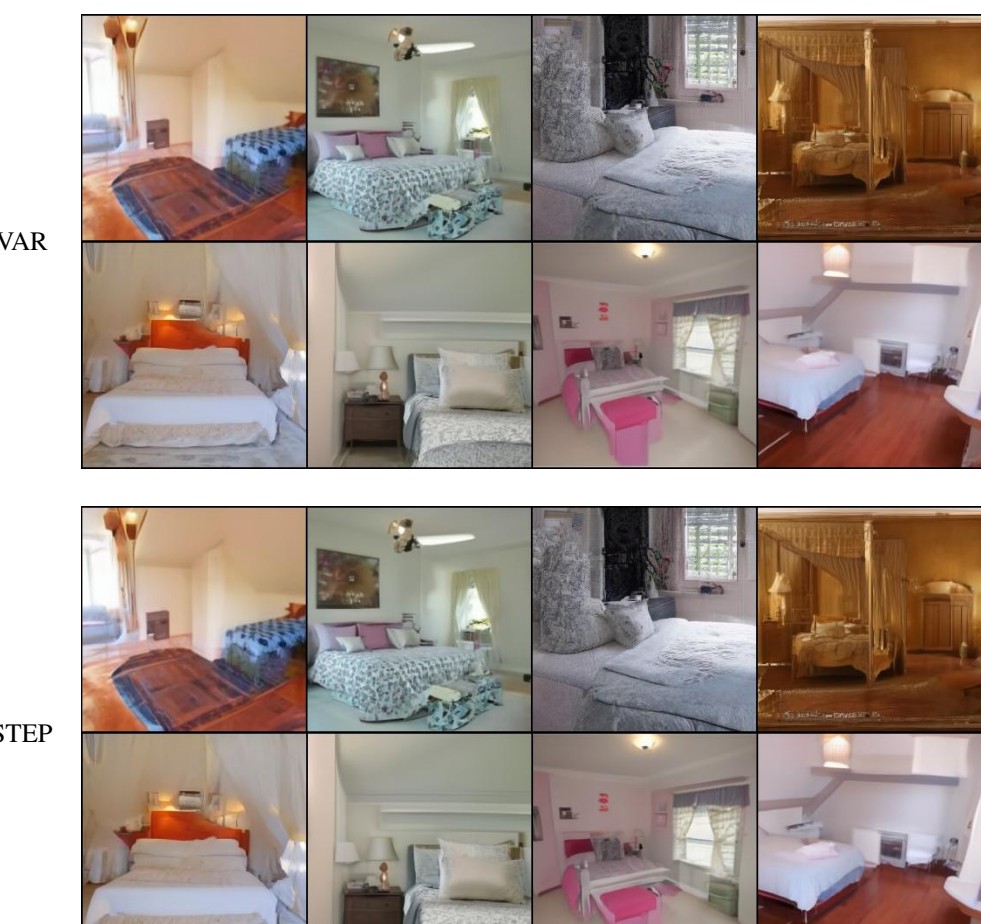

*Figure 6.* Comparison of generated samples of FastDPM on LSUN bedroom among different approximate diffusion processes. The approximate reverse process is DDIM-rev ($\kappa = 0.0$) and $S = 20$.

VAR

STEP

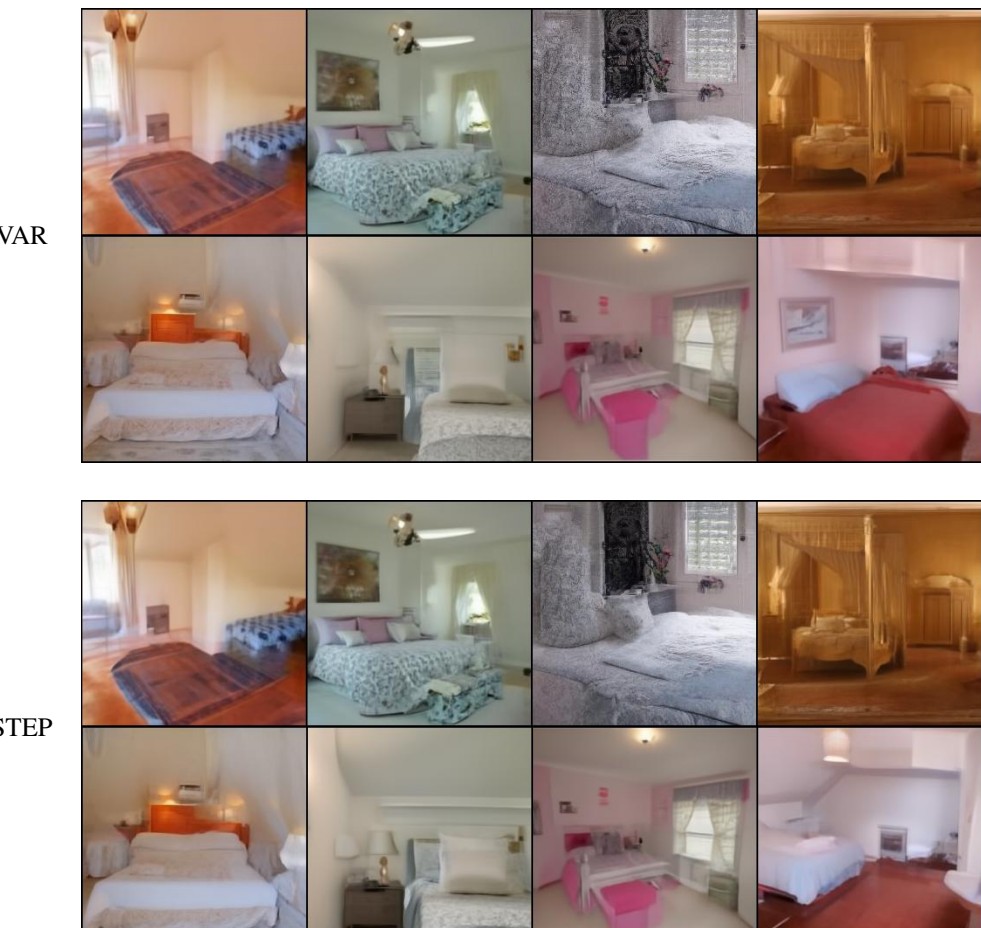

*Figure 7.* Comparison of generated samples of FastDPM on LSUN bedroom among different approximate diffusion processes. The approximate reverse process is DDIM-rev ($\kappa = 0.0$) and $S = 10$.

### E.4. Unconditional Generation on SC09

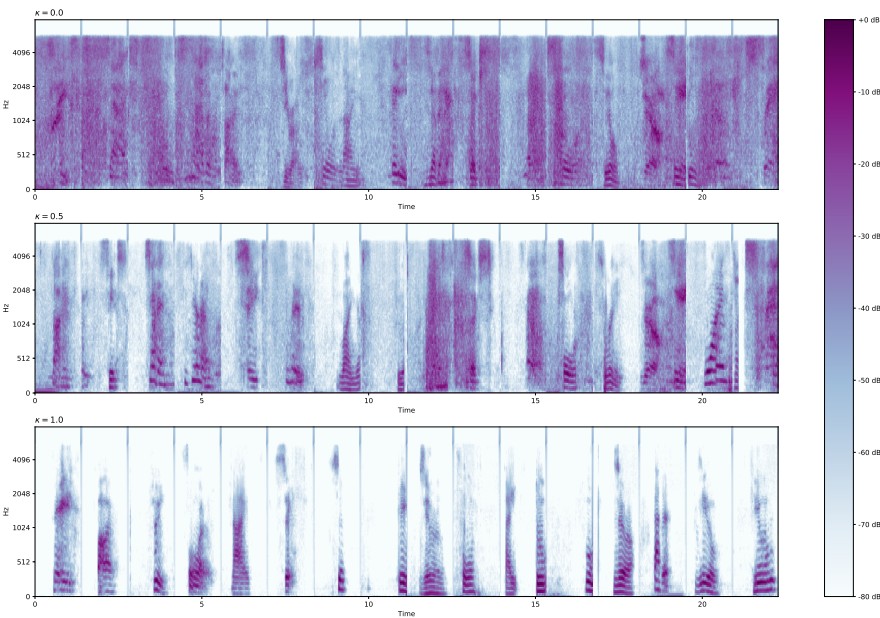

(a) STEP + DDIM-rev ($\kappa = 0.0$) (top) / DDIM-rev ($\kappa = 0.5$) (middle) / DDPM-rev (bottom)

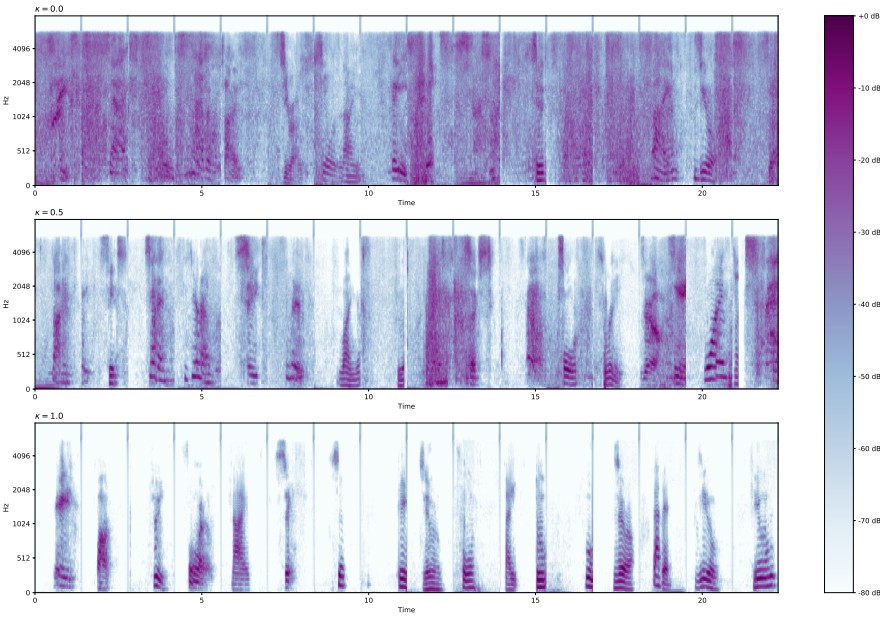

(b) VAR + DDIM-rev ($\kappa = 0.0$) (top) / DDIM-rev ($\kappa = 0.5$) (middle) / DDPM-rev (bottom)

*Figure 8.* Mel-spectrogram of 16 synthesized utterances ($S = 50$). We use linear noise level schedules from steps in (a) and variances in (b). In each subplot, the top row shows results of DDIM-rev ($\kappa = 0.0$), the middle row shows results of DDIM-rev ($\kappa = 0.5$), and the bottom row shows results of DDPM-rev. DDPM-rev produces the clearest utterances in these approximate reverse processes.

### E.5. Conditional Generation on SC09

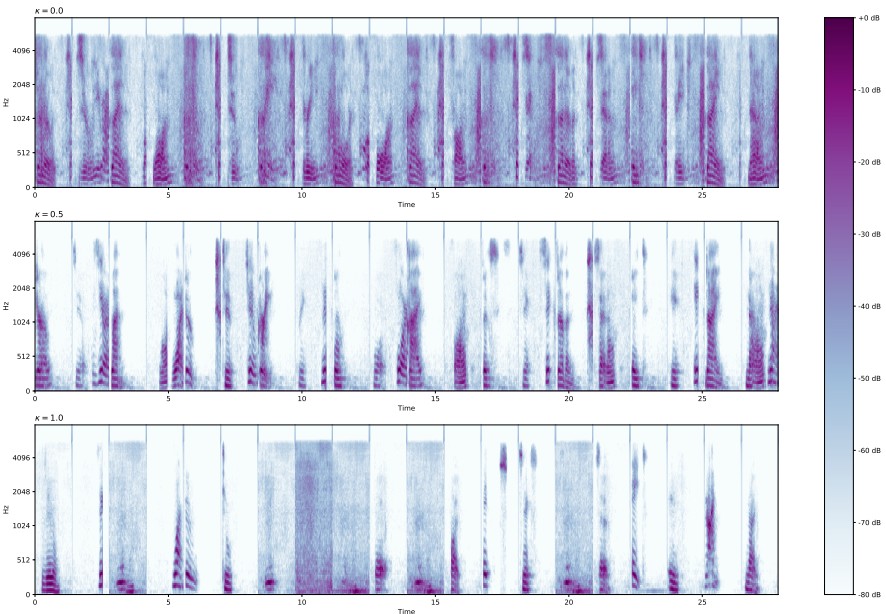

(a) STEP + DDIM-rev ($\kappa = 0.0$) (top) / DDIM-rev ($\kappa = 0.5$) (middle) / DDPM-rev (bottom)

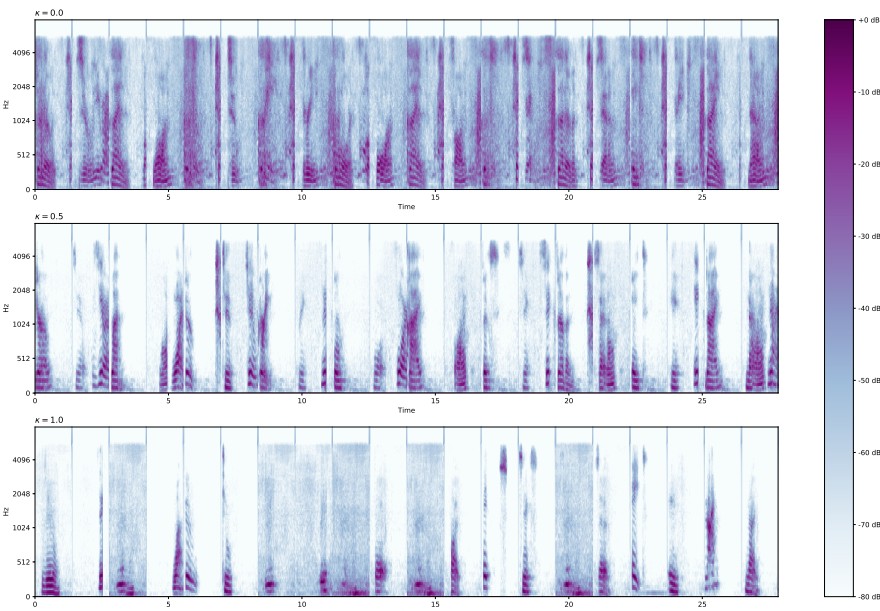

(b) VAR + DDIM-rev ($\kappa = 0.0$) (top) / DDIM-rev ($\kappa = 0.5$) (middle) / DDPM-rev (bottom)

*Figure 9.* Mel-spectrogram of 20 synthesized utterances ($S = 50$). We use linear noise level schedules from steps in (a) and variances in (b). In each subplot, the top row shows results of DDIM-rev ($\kappa = 0.0$), the middle row shows results of DDIM-rev ($\kappa = 0.5$), and the bottom row shows results of DDPM-rev. DDIM-rev ($\kappa = 0.5$) produces the clearest utterances in these approximate reverse processes.

## E.6. Neural Vocoding on LJSpeech

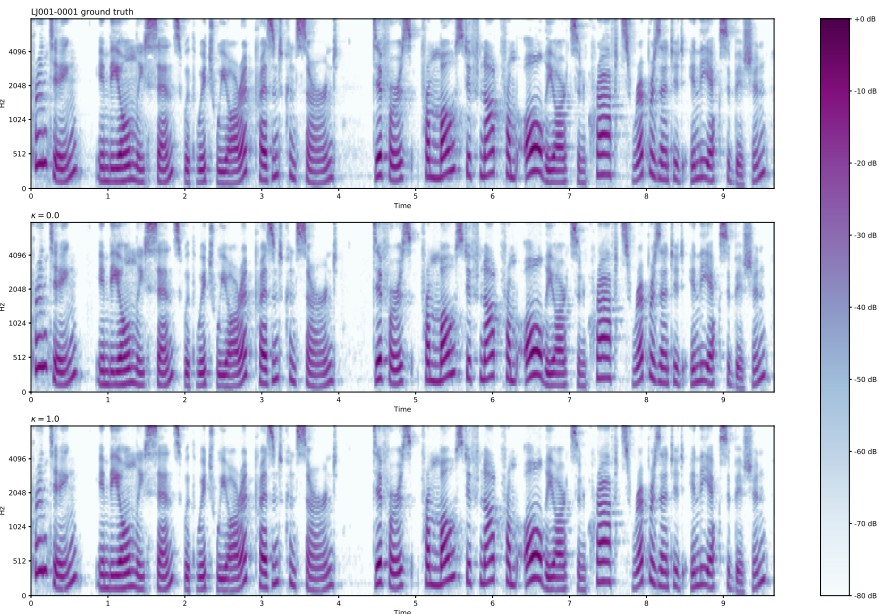

(a) Ground truth (top) / STEP + DDIM-rev (middle) / STEP + DDPM-rev (bottom)

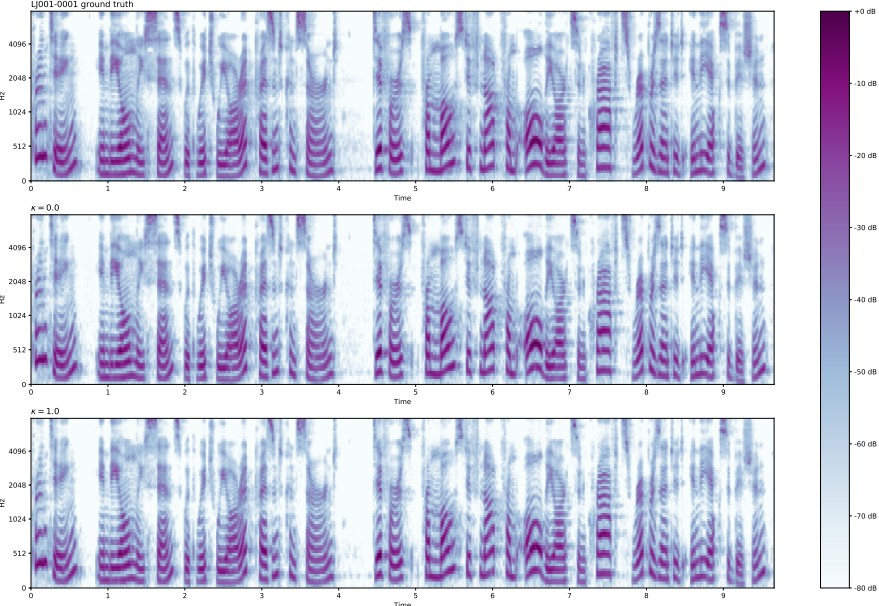

(b) Ground truth (top) / VAR + DDIM-rev (middle) / VAR + DDPM-rev (bottom)

*Figure 10.* Mel-spectrogram of ground truth and generated LJ001-0001 ($S = 5$, channel$= 128$). We use linear noise level schedules from steps in (a) and variances in (b). In each subplot, the top row shows ground truth, the middle row shows results of DDIM-rev ($\kappa = 0.0$), and the bottom row shows results of DDPM-rev. Both DDPM-rev and DDIM-rev generate high quality speech.