# OpenReview forum: "On Fast Sampling of Diffusion Probabilistic Models"
_ICML.cc/2021/Workshop/INNF — INNF+ 2021 spotlighttalk_

### Official Review · Reviewer_z9Wx · 2021-06-11

**Rating:** Accept
**Confidence:** 3

**Summary:**

Diffusion-based models are gaining traction currently due to their flexibility and scalability, combining the best traits of adversarial and likelihood-based generative models. One area where they are still more expensive than most other generative modelling paradigms (perhaps excepting autoregressive likelihood-based models) is sampling, and this paper is a timely exploration of what can be done to improve this situation.

In particular, the paper bridges the gap between diffusion-based models where time has been discretised up front (e.g. DDPM), and continuous-time diffusion-based models, where discretisation can be done adaptively during sampling (e.g. the SDE-based paradigm of Song et al. 2020b). The proposed approach makes the trade-off between speed of sampling and sample quality more accessible for discrete-time diffusion-based models.

**Justification For Rating:**

Endowing discrete-time models with this capability is valuable, though it did make me wonder if we shouldn't just be sticking to continuous time instead, where all of this seems to come more naturally, and where we can rely on existing literature on stochastic differential equations to find better/faster sampling procedures. That said, as argued in the paper, the current state-of-the-art results with diffusion models are held by discrete-time models, so from that point of view it makes sense to focus on this variant (at least while it has not been shown that continuous-time models are able to match these SoTA results).

One of the most interesting findings of this work is that there is no one-size-fits-all approach, and which method performs best depends on the nature of the data (concretely, images vs. audio). It is somewhat disappointing that the no-free-lunch theorem also holds in this context, but it is definitely valuable to know that this is the case.

It took me a bit of time to understand where the Gamma function in (3) comes from (it seemed arbitrary at first), but the terseness of this derivation is probably down to the page limit, given that the experimental results are all in the appendices. In a longer version of this work, including a more detailed derivation would definitely make it easier to understand.

I do quite appreciate section 4.1 as a concise summary of the empirical findings.

---

### Official Review · Reviewer_pyYc · 2021-06-12

**Rating:** Accept
**Confidence:** 2

**Summary:**

The paper proposes a framework for faster sampling in diffusion generative models without model retraining. The method is approximating pretrained Denoising Diffusion Probabilistic Model via shorter diffusion and reverse processes. The idea is based on generalizing discrete diffusion steps to continuous diffusion steps and providing an intertible mapping between diffusion steps and noise levels. They compare the proposed approximations to diffusion processes on a range of image generation tasks.

**Justification For Rating:**

The paper proposes a novel approach for fast sampling in diffusion models. The authors propose an interesting idea which can give rise to further research in this area, and it is relevant to the workshop. Generally, the paper it well written and easy to follow.
In the experiments section, I would encourage the authors to not only describe the results but also give more explanations and intuitions for the observed model's performance.

---

### Decision · Program_Chairs · 2021-06-14

Accept (spotlight talk)